# Mathematical modeling of the evolution of resistance and aggressiveness of high-grade serous ovarian cancer from patient CA-125 time series

Kanyarat Jitmana[1], Jason I. Griffiths[2], Sian Fereday[3,4], Anna DeFazio[5,6,7], David Bowtell[3,4], for Australian Ovarian Cancer Study, Frederick R. Adler[1,8,9] *

1 Department of Mathematics, The University of Utah, Salt Lake City, Utah, The United States of America, 2 Department of Medical Oncology and Therapeutics Research, City of Hope National Medical Center, Duarte, California, The United States of America, 3 Peter MacCallum Cancer Centre, Melbourne, Victoria, Australia, 4 Sir Peter MacCallum Department of Oncology, The University of Melbourne, Parkville, Victoria, Australia, 5 Centre for Cancer Research, The Westmead Institute for Medical Research, Sydney, New South Wales, Australia, 6 Department of Gynaecological Oncology, Westmead Hospital, Sydney, New South Wales, Australia, 7 The Daffodil Centre, The University of Sydney, a joint venture with Cancer Council NSW, Sydney, New South Wales, Australia, 8 School of Biological Sciences, The University of Utah, Salt Lake City, Utah, The United States of America, 9 Huntsman Cancer Institute, The University of Utah, Salt Lake City, Utah, The United States of America

* adler@math.utah.edu

**Data Availability Statement:** The data analyzed in this study are available upon reasonable request for

## Abstract

A time-series analysis of serum Cancer Antigen 125 (CA-125) levels was performed in 791 patients with high-grade serous ovarian cancer (HGSOC) from the Australian Ovarian Cancer Study to evaluate the development of chemoresistance and response to therapy. To investigate chemoresistance and better predict the treatment effectiveness, we examined two traits: *resistance* (defined as the rate of CA-125 change when patients were treated with therapy) and *aggressiveness* (defined as the rate of CA-125 change when patients were not treated). We found that as the number of treatment lines increases, the data-based resistance increases (a decreased rate of CA-125 decay). We use mathematical models of two distinct cancer cell types, treatment-sensitive cells and treatment-resistant cells, to estimate the values and evolution of the two traits in individual patients. By fitting to individual patient HGSOC data, our models successfully capture the dynamics of the CA-125 level. The parameters estimated from the mathematical models show that patients with inferred low growth rates of treatment-sensitive cells and treatment-resistant cells (low model-estimated aggressiveness) and a high death rate of treatment-resistant cells (low model-estimated resistance) have longer survival time after completing their second-line of therapy. These findings show that mathematical models can characterize the degree of *resistance* and *aggressiveness* in individual patients, which improves our understanding of chemoresistance development and could predict treatment effectiveness in HGSOC patients.

academic purposes only due to data contain potentially identifying or sensitive patient information. Data can be access by application to the independent Peter Mac Data Access Committee at dac@petermac.org.

**Funding:** This research was supported in part by City of Hope, NIH-CSBC: U54 CA209978 to FRA. KJ received support from the Modeling the Dynamics of Life fund at the University of Utah. Australian Ovarian Cancer Study was supported by the U.S. Army Medical Research and Materiel Command (DAMD17-01-1-0729), NH\&MRC of Australia (199600, 400413, and 400281), Cancer Councils of NSW, Victoria, Queensland, South Australia and Tasmania and Cancer Foundation of Western Australia (191, 211 and 182). Australian Ovarian Cancer Study gratefully acknowledges additional support from Ovarian Cancer Australia and the Peter MacCallum Foundation. The funders had no role in study design, data analysis, decision to publish, or preparation of the manuscript.

**Competing interests:** The authors declare that they have NO affiliations with or involvement in any organization or entity with any financial interest in the subject matter or materials discussed in this manuscript. SF and AD declare receiving a research grant from AstraZeneca, and DB research grant funding from AstraZeneca and Genentech Roche and is an advisor to Exo Therapeutics for work not directly related to the analyses described in this manuscript.

## Author summary

Ovarian cancer is a major cause of death in women worldwide due to the emergence of treatment resistance and eventual treatment failure. Serum levels of the biomarker Cancer Antigen-125 (CA-125) can be used to monitor treatment response in patients with epithelial ovarian cancer. We used time series of CA-125 in 791 patients with high-grade serous ovarian cancer (HGSOC) from the Australian Ovarian Cancer Study to quantify the evolution of resistance and aggressiveness as a response to therapy in individual patients to predict the dynamics of CA-125 and the survival outcomes. We present two mathematical models that include treatment-resistant cells and treatment-sensitive cells. These models accurately fit the data and characterize patients with the best outcomes as those with the least model-estimated aggressively growing cells and the least model-estimated resistant cells. Models with only a single cell type provide poor fits to the data. These minimal models with just two cell types could provide a valuable tool for rapidly and robustly understanding the dynamics of individual patients and pointing the way to identifying specific mechanisms.

## Introduction

Mathematical models have been widely used to provide valuable insights into biological processes and cancer biology, offering a quantitative framework to study and analyze the complex dynamics of cancer resistance. In cancer studies, models can help analyze large amounts of patient data to identify risk factors, develop personalized treatment plans, and monitor disease progression [1–3]. Ovarian cancer is a fatal gynecological malignancy and ranks fifth among the deadliest cancers in women worldwide. Around 1.2% of women are estimated to be diagnosed with ovarian cancer during their lifetime [4–6], making it the eighth most prevalent cancer in women [7]. Older women have a higher chance of developing ovarian cancer; more than 50% of ovarian cancer cases occur in women over 60 [8]. Despite its name, recent studies have shown that ovarian cancer does not always originate in the ovaries but can begin in other pelvic organs, such as the fallopian tubes, and invade the ovaries secondarily [9–11]. High-grade serous ovarian cancer (HGSOC) is the most common histological subtype of epithelial ovarian cancer. It is characterized in almost all cases by somatic *TP53* alterations, and defects in homologous recombination DNA repair genes occur in approximately 50% of HGSOC cases, including alterations in *BRCA1* and *BRCA2* [12, 13].

The primary treatment for HGSOC patients typically consists of cytoreductive surgery and either adjuvant or neo-adjuvant chemotherapy, most commonly a combination of carboplatin and paclitaxel. Contemporary treatment includes maintenance poly (ADP-ribose) polymerase inhibitors (PARPi) with or without bevacizumab in a selected subset of patients [14]. Subsequent lines of treatment are chosen largely based on the patient's response to the previous treatment line, particularly the time between the end of carboplatin-based chemotherapy and subsequent relapse, known as the platinum-free interval [15]. In patients with HGSOC, serum Cancer Antigen-125 (CA-125) levels are widely used to monitor the treatment response [7, 16]. CA-125 is expressed on the surface of epithelial ovarian cancer cells. It is shed into the circulation and can be quantified in serum [6, 16, 17]. CA-125 levels differ across ovarian cancer subtypes, with HGSOC showing the highest average expression [16, 18]. Following surgical intervention and/or chemotherapy, a reduction in CA-125 levels in the bloodstream is expected as a marker of treatment effectiveness. Conversely, an elevation in CA-125 levels may suggest either inadequate therapeutic response or disease progression. In cases where the

primary treatment for ovarian cancer involves a combination of surgery and chemotherapy, which is usually the standard of care, the decrease in CA-125 levels may not directly indicate the response to chemotherapy because the decline in CA-125 levels is a result of both treatment modalities working in tandem [19]. During their cancer journey, women with ovarian cancer undergo periodic CA-125 blood tests as a standard protocol for monitoring. An increase in the levels of CA-125 can indicate cancer recurrence and progression, which would require further diagnostic evaluations, such as imaging studies, to confirm disease progression [20]. Nonetheless, monitoring CA-125 levels during first-line therapy helps predict chemotherapy sensitivity, resistance, survival, and progression-free interval [21]. Cancer recurrence and progression may indicate resistance to treatment.

After primary surgery and the first line of chemotherapy, approximately 70–80% of HGSOC ovarian cancer patients experience recurrence and require additional lines of therapy [22]. Patients with ovarian cancer are classified as platinum-resistant or platinum-sensitive based on their progression-free interval, which is often calculated from the end of the first line of treatment to the date of first progression or recurrence [15, 23]. With recurrent disease and multiple lines of therapy, resistance to therapy can evolve, leading to reduced response to treatment [24, 25]. Many mechanisms that underlie chemoresistance evolution in ovarian cancer have been identified [26]. In the absence of detailed cell-specific data at each point of treatment failure, predicting the response to subsequent treatments is difficult. Inferring the properties of cancer cells without detailed cell-specific data can be overcome by using mathematical models [27–31]. Mathematical models that simulate cancer cell behavior and interactions with the tumor microenvironment provide insights into tumor development, progression, therapy effectiveness, biomarkers, and drug resistance.

In this study, we analyzed the response to ovarian cancer treatment using the dynamics of CA-125 in HGSOC patients to predict overall survival time. We hypothesized that two cell properties characterize ovarian cancer patients: *resistance* and *aggressiveness. Resistance* describes cell population change when patients were treated during lines of treatment, and *aggressiveness* describes cells change when patients were not treated between treatment lines. Patients can differ in their initial level or frequency of resistance or in the rate at which resistance and aggressiveness evolve. Accordingly, we developed two contrasting models of heterogeneity based on serial CA-125 measurements in a large cohort of women with HGSOC. The first model uses ordinary differential equations (ODEs) with two types of cancer cells: drug-resistant and drug-sensitive. Models with drug sensitive and drug resistant cells have been well studied to understand the dynamics of Prostate-Specific Antigen (PSA) in prostate cancer [32–35]. Such models have not been applied to study CA-125 levels in ovarian cancer. A simple linear ODE model describing the replication and death of sensitive and resistant cells is easily understandable and interpretable, making it highly accessible. The model offers faster computation and provides insights into the basic dynamics of the cancer population. The second model builds on the framework of adaptive dynamics from evolutionary ecology to model the evolution of the average level of drug resistance in the cancer cell population [36–38]. The model provides a continuum of cell types, which can capture the variation of treatment-resistance levels in the cancer cell population.

Our overall aim is to use mathematical models of CA-125 dynamics to understand tumor evolution and evaluate factors that predict overall survival. Our approach leverages clinical data to identify the key factors and covariates impacting patient survival using statistical methods such as Cox proportional hazards. We investigate a simple mathematical model of *resistance* and *aggressiveness* to capture the dynamics of CA-125 levels in a large cohort of patients with HGSOC. We compare the predictive performance of an ODE model with separate treatment-sensitive and treatment-resistant cell populations against an adaptive dynamic model

with a continuum of levels of treatment-resistant levels to determine which model better captures the dynamics of CA-125 levels. By individually fitting the initial lines of therapy of each HGSOC patient from clinical data to a mathematical model, we investigate whether the models can accurately capture the short-term and long-term dynamics of CA-125 levels, predict its future dynamics, and enhance the accuracy of survivorship predictions.

## Materials and methods

### Ethics statement

All procedures performed in studies involving human participants were in accordance with the ethical standards of the institutional and/or national research committee and with the 1964 Helsinki Declaration and its later amendments or comparable ethical standards. Australian Ovarian Cancer Study was approved by the Human Research Ethics Committees of the Peter MacCallum Cancer Centre, and all participating centres and all Australian Ovarian Cancer Study participants provided written consent.

### Data analysis

**Data.** Patients with HGSOC were identified in the Australian Ovarian Cancer Study (AOCS), an Australia-wide population-based molecular epidemiological study [39]. The dataset includes longitudinal serum CA-125 level measurements from 1057 patients with HGSOC. Among these patients, 1052 (99.5%) underwent primary treatment that included either primary debulking surgery (PDS) or neoadjuvant chemotherapy (NACT). A total of 882 HGSOC patients (83.44%) progressed, and 793 HGSOC patients (75.02%) died during the study period (median survival of 3.6 years, with a range of 17 days to 33.9 years). To be considered for our statistical and mathematical analysis, HGSOC patients were required to have undergone second-line treatment after disease progression. Patients with fewer than six data points for CA-125 were excluded from the study. Of 1057 patients, a total of 791 patients met these qualification criteria and were included in the analysis. Additional details regarding the inclusion criteria can be found in S1 Fig.

In addition to patient-specific CA-125 levels, clinical data included the normal range of CA-125 levels, patient age at diagnosis, types of drugs in each line of treatment, the primary surgery date, residual disease following cytoreductive surgery, cause of death, and the date of the last follow-up. From the clinical data, we used linear regression to estimate the ***data-based resistance*** as the slope of the log-transformed CA-125 level change when patients were treated during each line of therapy and the ***data-based aggressiveness*** as the slope of the log-transformed CA-125 level change when patients were not receiving therapy in each individual patient. The values of data-based resistance and data-based aggressiveness serve as descriptors of changing in CA-125 level to show how patients respond during and between treatments. The estimation process was performed separately for each individual patient for each treatment line (Fig 1A). The data-based resistance and data-based aggressiveness were estimated using linear regression with patients treated as fixed effects (lm [40]) or random effects (lme4 [41] in *R* [40]). Fixed effects treat patients as independent individuals and random effects treat patients as members of a population.

**Statistical analysis.** All statistical analyses were performed using software R [40]. The survival time of HGSOC patients after each line of treatment was measured as the time between the end of the treatment line and the day of death or the last follow-up. The progression-free interval (PFI) after the first-line treatment was measured as the time between the end of the first line of chemotherapy and the date of disease progression. The Cox proportional hazards (coxph of the survival package [42] in *R*) was used to analyze survival times and PFI. The

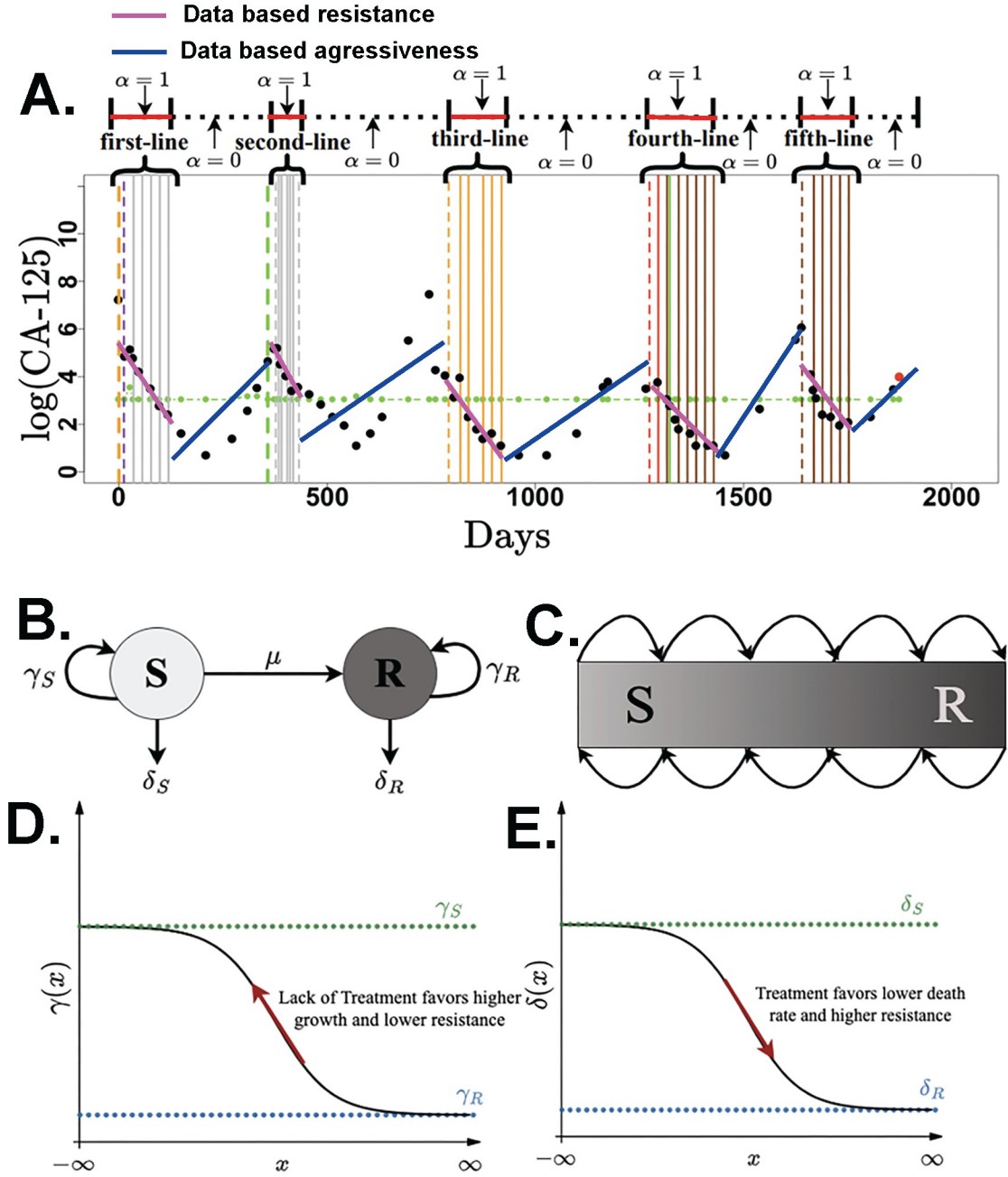

**Fig 1. Mathematical model diagram. A.** Scatter plot of the CA-125 level (black dots) on a log scale over time for a patient who underwent five lines of treatment. The value of $\alpha$ as defined in both the SR and AD models is shown. The horizontal green dots represent the upper limit of normal for CA-125. The vertical thick dashed orange and green lines show the date of diagnosis and the date of disease progression, respectively. The red dot indicates the date of death. Each individual vertical line denotes the date on which the patient received the treatment. Different colors of the vertical lines indicate the use of distinct drugs in each treatment regimen. First-line, second-line, third-line, fourth, and fifth-line indicate the line of treatment, during which $\alpha = 1$. We set $\alpha = 0$ between lines of treatment.**B.** The structure of the treatment-sensitive ($S$) and treatment-resistant ($R$) cell model (SR model). The parameters $\gamma$ and $\delta$ correspond to the rates of cell population growth between treatments and death during treatment, respectively. The parameter $\mu$ represents the rate of treatment-sensitive transformation to treatment-resistant cells. **C.** The structure of the adaptive dynamics model (AD model). Different shades of color denote the level of chemoresistance, with darker colors indicating a higher level of chemoresistance. The plot of the evolution of **D**. death rate and **E.** growth rate as a function of the chemoresistance level ($x$).

Kaplan-Meier plots are analyzed for significance using the log-rank test. In this study, we used the Wilcoxon test to the survival analysis and computed the p-value. We used the Pearson correlation though out the study to test the linear correlation between the variable of interest.

## Mathematical modeling

**Treatment-sensitive and treatment-resistant cell model (SR model).** The treatment-sensitive and treatment-resistant cells model (SR model) tracks the continuous-time dynamics of the populations of two types of cancer cells: treatment-sensitive cells ($S$) and treatment-resistant cells ($R$) (Fig 1**B**). During lines of treatment, cells die at rate $\delta_R$ for treatment-resistant cells and rate $\delta_S$ for treatment-sensitive cells. Similarly, the growth rate of cells when patients were not undergoing treatment are denoted by $\gamma_S$ for treatment-sensitive cells and $\gamma_R$ for treatment-resistant cells. The death rate ($\delta$) thus represents *resistance*, and the growth rate ($\gamma$) represents aggressiveness. Treatment-sensitive cells can become treatment-resistant cells at the transition rate $\mu$. Information of parameters can be found in Table 1. The assumptions regarding the death rate ($\delta$) and the growth rate ($\gamma$) for the sensitive cells and resistant cells are as follows:

1. In order to maintain model simplicity, all growth and death rates ($\gamma_R, \gamma_S, \delta_R, \delta_S$) remain constant over time, regardless of the distinct treatment regimens employed in each treatment line.

2. Both sensitive and resistant cells have the capacity to proliferate in the absence of treatment. It is assumed that the growth rates ($\gamma_S, \gamma_R$) are positive values.

3. To denote the impact of treatment on sensitive cells, the death rate of sensitive cells ($\delta_S$) is constrained to be positive.

4. Due to the limited efficacy of treatment on resistant cells, the death rate of resistant cells ($\delta_R$) can take on positive or negative values.

**Table 1. Model parameters.**

| Parameter Description | Parameter | Units |
|---|---|---|
| *Estimated parameters (all models):* | | |
| The aggressiveness or growth rate of resistant cells | $\gamma_R$ | per day |
| The aggressiveness or growth rate of sensitive cells | $\gamma_S$ | per day |
| The resistance or death rate of resistant cells | $\delta_R$ | per day |
| The resistance or death rate of sensitive cells | $\delta_S$ | per day |
| *Estimated parameters (SR model):* | | |
| The transition rate from sensitive to resistant cells | $\mu$ | per day |
| *Estimated parameters (AD model):* | | |
| Level of resistance setting the half-maximum growth rate | $A_\gamma$ | - |
| Level of resistance setting the half-maximum death rate | $A_\delta$ | - |
| Shape preserving of the curve representing growth rate | $B_\gamma$ | - |
| Shape preserving of the curve representing death rate | $B_\delta$ | - |
| *Parameters from data:* | | |
| The treatment status (1 when on treatment, 0 otherwise) | $\alpha$ | - |

Summary of parameters used in the models.

To link with the data, treatment-sensitive ($S$) and treatment-resistant cells ($R$) were assumed to proportionally generate amounts of CA-125 and neglected other CA-125 production sources. The level of CA-125 ($C$) is given by

$$C = R + S. \tag{1}$$

An extended model, where $C$ follows a differential equation with a half-life $\delta_C$,

$$\frac{dC}{dt} = \delta_C(R + S - C) \tag{2}$$

produces similar results.

The complete system of differential equations for cancer cell dynamics is

$$\frac{dR}{dt} = (1 - \alpha)\gamma_R R - \alpha\delta_R R + \mu S \tag{3}$$

$$\frac{dS}{dt} = (1 - \alpha)\gamma_S S - \alpha\delta_S S - \mu S \tag{4}$$

where $\alpha = 1$ when the patient is treated during lines of treatment and $\alpha = 0$ when the patient is off treatment (Fig 1A). In Fig 1A, the designations "first-line", "second-line", "third-line", etc., correspond to different treatment lines. During these lines of treatment, $\alpha = 1$ and it is 0, otherwise. The vertical lines in lines of treatment symbolize the specific instances when drugs are administered to the patient, which is called treatment cycle. The group of cycles including the gap between them is line of treatment.

The SR model was tested under three scenarios, providing mechanisms for the emergence of treatment-resistant cells assuming the value of initial condition of CA-125 ($C(0)$ is taken from clinical data. First, in the SR model with a pre-existing treatment-resistant cell population (the SR and R0 model), treatment-resistant cells are present in the initial conditions before treatment commenced, with parameters $\mu = 0$, $R(0)>0$, and $C(0) = R(0) + S(0)$. Second, the SR model with treatment-resistant cells arising from the transformation of treatment-sensitive cells (the SR and $\mu$ model), without the presence of initial treatment-resistant cells at the start of treatment, with specific parameters $\mu > 0$, $R(0) = 0$, and $C(0) = S(0)$. Third, the single cell-type model has no resistant cells, created as a special case of the SR model with parameters $\mu = 0$, $R(0) = 0$, and $C(0) = S(0)$.

**Adaptive dynamics model.** The adaptive dynamics model (AD model) considers a continuum of cell types instead of just two distinct cell types as in the SR model [43] (Fig 1C). At any given time $t$, it is assumed that all cancer cells ($N$) have the same level of chemoresistance ($x$). The cancer cell population has a death rate $\delta(x)$ during treatment ($\alpha = 1$) and a growth rate $\gamma(x)$ when patients were not receiving treatment ($\alpha = 0$). The level of chemoresistance ($x$) evolves up the fitness gradient with the net growth of the population. The equations governing the adaptive dynamics model are

$$\frac{dN}{dt} = (1 - \alpha)\gamma(x)N - \alpha\delta(x)N \tag{5}$$

$$\frac{\partial x}{\partial t} = K\frac{\partial W}{\partial x}, \tag{6}$$

where $W(x) = (1 - \alpha)\gamma(x) - \alpha\delta(x)$ is the net growth of population $N$. The equation for $x$ describes the evolution, and the parameter $K$ sets the response rate to the selection gradient,

which is set to 0.01. The growth rate $\gamma(x)$ and death rate $\delta(x)$ are defined functions as

$$\gamma(x) \quad = \quad \gamma_R + (\gamma_S - \gamma_R)\frac{1}{1 + e^{B_\gamma x + A_\gamma}} \tag{7}$$

$$\delta(x) \quad = \quad \delta_R + (\delta_S - \delta_R)\frac{1}{1 + e^{B_\delta x + A_\delta}}. \tag{8}$$

As $x \to \infty$ the cancer cell population is made up of the most treatment-resistant cells with growth rate $\gamma_R$. Conversely, as $x \to -\infty$, the cancer cell population consists of the most treatment-sensitive cells with a growth rate $\gamma_S$ and a death rate $\delta_S$. Treatment thus favors a lower death rate and increased chemoresistance ($x$) (Fig 1**D**), and lack of treatment favors a higher growth rate and decreased chemoresistance ($x$) (Fig 1**E**). The assumptions regarding $\delta_S$, $\delta_R$, $\gamma_S$, and $\gamma_R$ are the same as in the SRmodel. Information of parameters can be found in Table 1.

**Parameter estimation.** We used minimization of least squares to estimate the model parameters for each of the 791 HGSOC patients who had completed at least two lines of therapy and fulfilled the applicable data criteria. The models were fitted to clinical data using four different sets of data, including all available data points (*all-line fitting*), data from the two initial lines (*two-line fitting*), data from the three initial lines (*three-line fitting*), and data from the first four lines of treatment (*four-line fitting*). The sum of the squared deviations between the log-transformed CA-125 and the values predicted by each model was minimized. With a model written as [44]

$$\frac{d\mathbf{Z}}{dt} = f(\mathbf{Z}, \mathbf{t}, \theta), \tag{9}$$

where $\theta$ represents the set of parameters of the model, $\mathbf{Z} = \mathbf{Z}(\mathbf{t}, \theta)$ is the state variables, and $t$ is the time variable. We estimated the parameters by minimizing the objective function,

$$\sum_{t_i} ||\mathbf{Y_i} - \mathbf{Z_i}(\mathbf{t_i}, \theta)||_2^2, \tag{10}$$

where $t_i$ represents the time with measurements and $\mathbf{Y_i}$ the observed data at a time $t_i$. We use the function *optim* [40] in the software R to minimize the sum of squared errors and estimate the parameters $\theta$. For both the SR and AD models, the rate of growth of treatment-sensitive cells, the rate of growth of treatment-resistant cells, the death rate of treatment-sensitive cells, the rate of treatment-sensitive cells transform to treatment-resistant cells, and the initial value of treatment-resistant cells were estimated under the constraint that $\delta_S \geq 0$, $\gamma_S \geq 0$, $\gamma_R \geq 0$, $R(0) \geq 0$, and $\mu \geq 0$. The model fitting and parameter estimation are carried out individually for each patient. The initial estimates for each parameter ($\delta_R$, $\delta_S$, $\gamma_R$, and $\gamma_S$) are derived from data-based measurements of resistance and aggressiveness.

To explore whether the early response can predict later dynamics of CA-125 level, the parameters estimated with *two-line fitting*, *three-line fitting*, and *four-line fitting* were used to numerically solve the equations using the actual dates of treatment for that patient (S2 Fig).

**Model comparison.** The estimated parameters were utilized to numerically solve the models for all 791 eligible HGSOC patients, which were then compared to the observed CA-125 levels in the clinical data. To compare and determine the effectiveness of the SR and the AD models with *all-line fitting*, we compared the model's predicted CA-125 levels against the prediction of a smoothing function applied to the data (Friedman's SuperSmoother) [40] because of the large number of data points for each patient and the variability in the data. In this context, the Friedman's SuperSmoother function is a reference point that provides a baseline for assessing any model's ability to faithfully replicate the main trends in the data. The

goodness of fit ratio is given by:

$$\text{ratio} = \frac{\text{RSS}_{\text{sup}}}{\text{RSS}_{\text{model}}}, \tag{11}$$

where $\text{RSS}_{\text{model}}$ is the sum of the squared errors between the clinical CA-125 values and the CA-125 levels predicted by the model. The $\text{RSS}_{\text{sup}}$ represents the sum of the squared deviations between the clinical CA-125 values and the CA-125 levels predicted by the *supsmu* function. Specifically, when this ratio close to 1, the fit of the model to the observed data aligns closely to the fit of the SuperSmoother function. Conversely, deviations of this ratio from 1 quantify discrepancies between the model's predictions and the empirical data in comparison with the SuperSmoother function. We use the coefficient of determination, $R^2$, to assess the power of the mathematical model to explain the observed data. For model comparison, we use the Akaike Information Criterion (AIC) which favors models with a good fit to the data but penalizes additional parameters. A lower AIC supports a model that better balances fit and simplicity, and is given by

$$\text{AIC} = 2k - 2\ln L, \tag{12}$$

where $k$ is number of parameters in the model, $\ln L$ is log likelihood of the model [45]. When comparing *two-line fitting*, *three-line fitting*, and *four-line fitting*, the coefficient of determination ($R^2$) highlights the degree to which the model fits the clinical data. Given the smaller dataset for comparison and its familiarity we use $R^2$ to assess and compare the performance of *two-line fitting*, *three-line fitting*, and *four-line fitting*. Three distinct $R^2$ values were computed to assess the model's fitting and prediction of future CA-125 dynamics over the short and long run from the fitting lines. The goodness of fit of the model fits, as represented by ($R^2$ fitted line), is determined by calculating the $R^2$ value based on the levels of CA-125 used in the fitting process. The short-term prediction ($R^2$ next line) employs the CA-125 values from the subsequent line after the fitted lines to compute the $R^2$ value (for the *two-line fitting*, data from the third line of treatment is used for calculation). For long-term prediction ($R^2$ all later lines), all the CA-125 values not utilized in the fitting process are employed to compute the $R^2$ value. The graphical depiction of the data utilized for computing both $R^2$ values can be found in S2 Fig. The parameters ($\delta_R$, $\delta_S$, $\gamma_R$, $\gamma_S$, $R(0)$, and $\mu$) derived from fitting both the SR and AD models with the clinical data were utilized to evaluate the models' predictive capability for patient survival via survival analysis either in isolation or in combination with the estimates of data-based resistance and data-based aggressiveness directly from the data. The overall survival time is calculated from the end of the second-line treatment to the last follow-up or mortality date to avoid survivorship bias. Kaplan-Meier curves and Cox proportional hazards regression analysis were used to analyze the predictive value of these parameters.

## Results

### Statistical analysis

**Study selection and characteristics.**   A total of 791 patients diagnosed with High-Grade Serous Ovarian Cancer (HGSOC) were deemed eligible for inclusion in the study based on specific criteria, including disease progression, completion of second-line treatment, and the collection of more than six CA-125 data points and receiving a platinum-based regimen as their first-line therapy (S1 Fig). A Cox proportional hazards regression was performed to analyze the key factors that predict patients' survival. The overall survival time is calculated from the end of the second-line treatment to the last follow-up or mortality date. In terms of the first-line treatment, 633 out of 791 patients (80.03%) underwent primary debulking surgery

**Table 2. Cox proportional hazards regression.**

| Variables | n(%) | Hazard ratio with 95% CI | p-value |
|---|---|---|---|
| Residual disease | | | $5.72 \times 10^{-6}$ |
| -$\leq$1 cm | 185 (27.66%) | reference | |
| ->1 cm | 484 (72.34%) | 1.417 [1.219, 1.647] | |
| log(Pre-treatment CA-125) | | 1.448 [1.254, 1.673] | $4.93 \times 10^{-7}$ |
| Treatment in first-line | | | $1.09 \times 10^{-7}$ |
| -NACT | 158(29.97%) | reference | |
| -PDS | 663(80.03%) | 0.764 [0.634, 0.920] | |
| Age at diagnosis (years) | | 1.015 [1.007,1.023] | 0.00019 |
| PFI (days) | | 0.999[0.998, 0.9993] | $<2 \times 10^{-16}$ |

Adjusted hazard ratios (HRs) with 95% confidence interval of death after the second line of treatment for each variable using univariate Cox proportional hazards regression. log(Pre-treatment CA-125) represents the value of CA-125 before starting the first-line treatment in the log scale. The PDS is primary surgery followed by chemotherapy treatment, while NACT is the neoadjuvant chemotherapy.

followed by chemotherapy treatment (PDS), while 158 patients (29.97%) received neoadjuvant chemotherapy (NACT). The use of platinum drugs in the second line of treatment is generally based on the response to the first line by considering the duration of the progression-free interval (PFI; Methods). Patients with PFI greater than six months are usually categorized as platinum-sensitive and are more likely to receive platinum-based chemotherapy as the second-line treatment.

The data categorize patients into two groups based on the size of their residual disease following cytoreductive surgery: patients with residual disease limited to $\leq$1 cm maximum diameter and patients with residual disease >1 cm diameter. Patients with residual disease of $\leq$1 cm maximum diameter exhibited a statistically significant higher survival than patients with residual disease >1 cm diameter (Hazard ratio = 1.417 and Table 2). The use of PDS in the first-line treatment and lower diagnosis age was associated with a reduced risk of death after finishing the second-line treatment (Table 2). Patients with lower pre-treatment CA-125 levels demonstrated a longer overall survival period (Hazard ratio = 1.448 and Table 2). Patients with a longer progression-free interval (PFI) exhibited a higher likelihood of survival following the completion of second-line treatment (Hazard ratio 0.999 and $p < 2 \times 10^{-16}$).

**Changes in data-based resistance and data-based aggressiveness during lines of therapy.** *Data-based resistance* was defined as the slope of the change in log-transformed CA-125 levels during each line of therapy (when patients were on-treatment). *Data-based aggressiveness* was formally defined as the slope of the increase in log-transformed CA-125 levels during the interval from the conclusion of one line of treatment to the initiation of the subsequent treatment line. These values were estimated as a slope of CA-125 changing over time using linear regression with random effects across each individual line of the first six lines of therapy, separately. Because the data-based resistance is typically negative, a lower data-based resistance value indicates a faster decline in CA-125 level. An increase in CA-125 during lines of treatment indicates a high level of resistance. We estimate data-based resistance independently for each of the initial six lines of treatment. Data-based resistance increased as patients advanced through subsequent lines of treatment reflecting a diminishing treatment response after more treatment lines. (Fig 2A). Across patients, several correlations characterized different states. Patients with a high initial CA-125 level before first-line treatment showed a faster rate of CA-125 decrease during first-line treatment ($p < 2 \times 10^{-16}$ and S1 Table) but had no effect on the subsequent relapse. In contrast, data-based aggressiveness showed no clear trend

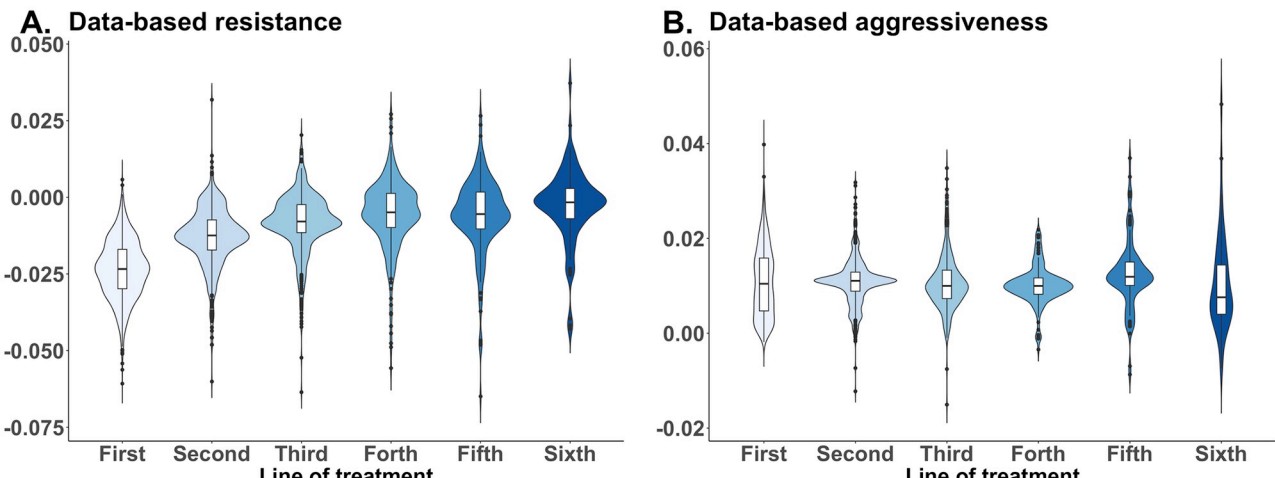

**Fig 2. Changes in data-based resistance and data-based aggressiveness during therapy. A**. Violin plot of the rate of CA-125 change when patients were treated (data-based resistance) against the lines of treatment in the x-axis. Each line was measured separately (i.e., the plot of line 2 is a rate of change of CA-125 during the second line of treatment). The value of data-based resistance increased as the number of lines increased (i.e., the tumor shrinkage became less negative), **B**. The rate of CA-125 increased when patients were off treatment (data-based aggressiveness) as a function of the number of lines of treatment. The number of lines in the x-axis indicates the last given line before the data-based aggressiveness is calculated. For example, the value for line 2 is calculated from the rate of change of the CA-125 level starting from the end of the second line to the beginning of the third line.

across different lines of treatment (Fig 2**B**). Patients with higher data-based resistance to first-line treatment showed weak but significantly higher data-based resistance to second-line treatment ($p = 0.012$ and S1 Table) and weakly but significantly lower data-based aggressiveness after first-line treatment ($p = 0.03$). Data-based aggressiveness after first-line treatment was correlated with data-based aggressiveness after second-line treatment ($p < 0.05$ and S2 Table). The correlations between data-based resistance and data-based aggressiveness became successively weaker with additional lines of treatment. The dosage is typically administered every 21 days. The duration of each treatment line ranges from 0 to 2,702 days, with a median duration of 106 days. The interval between successive treatment lines ranges from 0 to 4,879 days, with a median interval of 165 days.

We conducted a multivariate Cox proportional hazard analysis to examine if initial data-based resistance and data-based aggressiveness predict patient survival after completing the second-line treatment. Patients who exhibited lower data-based resistance during the first two lines of treatment and those with lower data-based aggressiveness after the first line of treatment had better survival rates (Fig 3 and S3 Fig).

## Mathematical modeling

The time course data for CA-125 reveal the presence of three broad dynamic patterns. In the first pattern, CA-125 fluctuates with treatments, decreasing during each line of therapy and increasing between lines. In the second pattern, CA-125 decreases during the initial treatment and increases thereafter, with the initial decrease likely because of the effects of surgery but with a lack of response to chemotherapy due to primary resistance. In the third pattern, CA-125 levels remain low after the initial line of treatment, which may indicate the presence of highly chemosensitive tumors and/or a long-lasting effect of surgery (S4 Fig).

**Fitting mathematical models to clinical data using *all-line fitting*.** When applying *all-line fitting* to the 791 HGSOC patients, the SR model with initial treatment-resistant cells (the

| Variable | N | Hazard ratio | | p |
|---|---|---|---|---|
| Data based-resistance in first line | 422 | | 0.87 (0.79, 0.96) | 0.008 |
| Data based-resistance in second line | 422 | | 0.92 (0.83, 1.04) | 0.175 |
| Data based-aggressiveness after first line | 422 | | 1.47 (1.35, 1.59) | <0.001 |

0.8　1　1.2 1.4

**Fig 3. Cox proportional hazards of data-based resistance and data-based aggressiveness.** Adjusted hazard ratios (HRs) with 95% confidence interval (CI) of time starting from the end of the second line of treatment using multivariate Cox proportional hazards regression analysis.

SR and R0 model), the SR model with the transition from treatment-sensitive cells to treatment-resistant cells (the SR and $\mu$ model), and the adaptive dynamics model (the AD model) capture the dynamics of CA-125 levels equally well when all data points were incorporated into the fitting process. (S4 Fig). Compared with the smoothed function using the ratio of sum of the square error (Eq 11), the goodness of fit of all three multiple-cell models (the SR and R0 model, the SR and $\mu$ model, and the AD model) are not significantly different from each other, and all are less than 1 ($p < 0.0001$). In contrast, the single cell-type model gave the worst fit to the clinical data. The ratio is 0.666±0.364 for the SR and $\mu$ model, 0.660±0.369 for the SR and R0 model, 0.713±0.388 for the adaptive dynamics model, and 0.238±0.239 for the single cell type (Fig 4A). We evaluated the goodness of fit using the $R^2$. The average $R^2$ values for the models are as follows: 0.834 ± 0.181 for the SR and $\mu$ model, 0.834 ± 0.194 for the SR and R0 model, 0.837 ± 0.196 for the Adp model, and 0.427 ± 0.774 for the Single cell model (Fig 4B). Similarly, the Akaike Information Criterion (AIC) values are consistent, showing no significant difference between the three models with two cell types (Fig 4C). Due to its inferior performance, we exclude the single-cell type model from further analysis. The current model operates under a fixed initial condition, $C(0)$. However, by treating the initial condition as an additional parameter, there is no notable impact on the model's fitting capabilities (S5 Fig).

The multiple-cell type models estimated the growth rate of treatment-sensitive ($\gamma_S$) and treatment-resistant cells ($\gamma_R$) in the absence of treatment, the death rate of treatment-sensitive cells ($\delta_S$) and treatment-resistant cells ($\delta_R$) during lines of treatment, and the initial population of treatment-resistant cells ($R(0)$) (Table 3). We estimate models parameters individually for each patient. Across individuals, the estimated growth and death rates are positively correlated for both sensitive ($\gamma_S$ and $\delta_S$) and resistant cells ($\gamma_R$ and $\delta_R$). This positive correlation suggests a trade-off between resistance and aggressiveness. We observe no significant correlation of the model-estimated growth rates of sensitive and resistant cells. The model-estimated initial population of treatment-resistant cells positively correlates with their associated death rate, the growth rate of sensitive cells, and the death rate of sensitive cells. Additionally, the model-estimated initial population of treatment-resistant cells negatively correlates with their growth

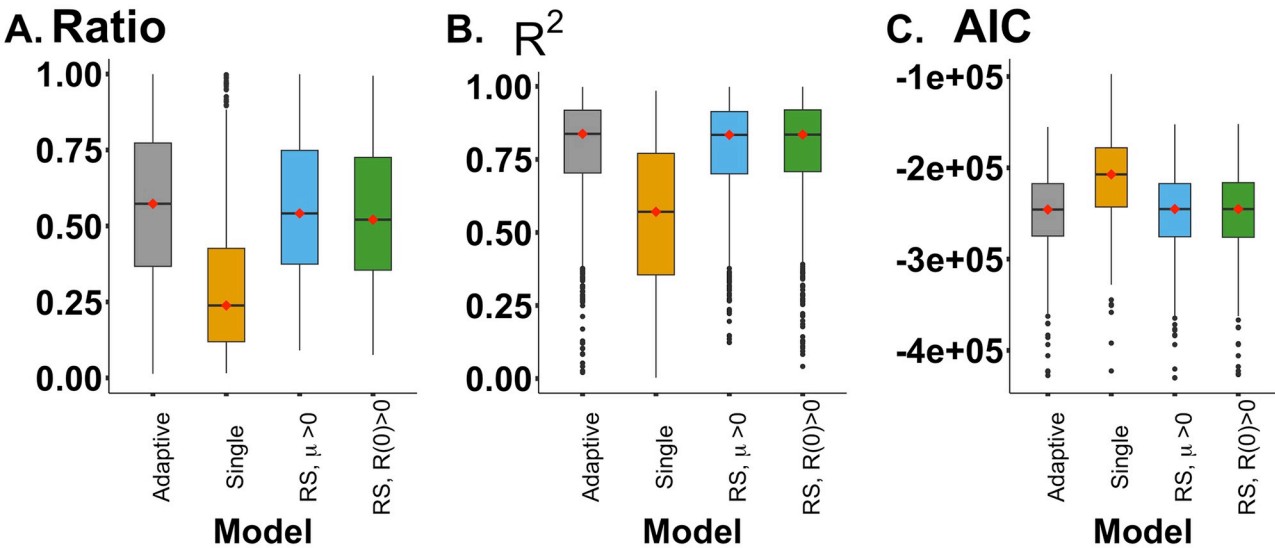

**Fig 4. The comparison of mathematical models. A.** Box plot of the ratio of the sum of the squared errors from the Friedman's SuperSmoother function to the sum of the squared errors from the mathematical models. **B.** Box plot of goodness of fit ($R^2$) using *all-lines fitting*. **C.** The Akaike Information Criterion (AIC) of the four models. The red dot indicates the median value of each variables.

rate. Similar results with the SR and $\mu$ model and the Adp model, with additional information on the correlation of parameters from the other models in the S6 Fig.

Parameter estimates across patients showed significant variance (Fig 5). We inferred the death rate of treatment-resistant cells to be significantly lower than the death rate of treatment-sensitive cells ($\delta_S > \delta_R$ and $p < 0.001$ with paired t-test). We found that the model-estimated aggressiveness was lower in treatment-resistant cells ($\gamma_S > \gamma_R$ with a $p < 0.0001$ with paired t-test and Fig 5**A**). The model-estimated resistance (death rate) of treatment-resistant cells was negative for 80% of all patients, indicating that these cells continued to grow during treatment (Fig 5**B**). However, this inequality holds for only 59% of patients ($p < 0.001$ with a chi-squared test).

We then assessed whether the estimated parameters could predict survivorship by categorizing patients into groups with parameter values above and below the median. Clinical covariates were not included in this initial analysis. Patients had more prolonged survival with less model-estimated aggressiveness in treatment-sensitive cells (small $\gamma_S$ increased median

**Table 3. Correlation of parameters.**

|  | $\delta_R$ | $\delta_S$ | $\gamma_S$ | $\gamma_R$ | $\log(R(0))$ |
|---|---|---|---|---|---|
| Mean | -0.00018 | 0.04421 | 0.0142 | 0.01024 | 0.1208 |
| s.d. | 0.0127 | 0.0241 | 0.0167 | 0.0127 | 0.0127 |
| range | (-0.0491,0.0431) | (0,0.0899) | (0,0.2441) | (0,0.0878) | (-5.5149,3.7757) |
| $\delta_R$ | - | *0.1038* | -0.0772 | **0.1438** | **0.1907** |
| $\delta_S$ |  | - | **0.2077** | **-0.1844** | **0.4221** |
| $\gamma_S$ |  |  | - | -0.0313 | **0.2213** |
| $\gamma_R$ |  |  |  | - | **-0.2289** |

Summary and correlations of parameter estimates across patients using the SR model with initial treatment-resistant cells. Italics indicates significance at $p < 0.05$ and bold $p < 0.0001$.

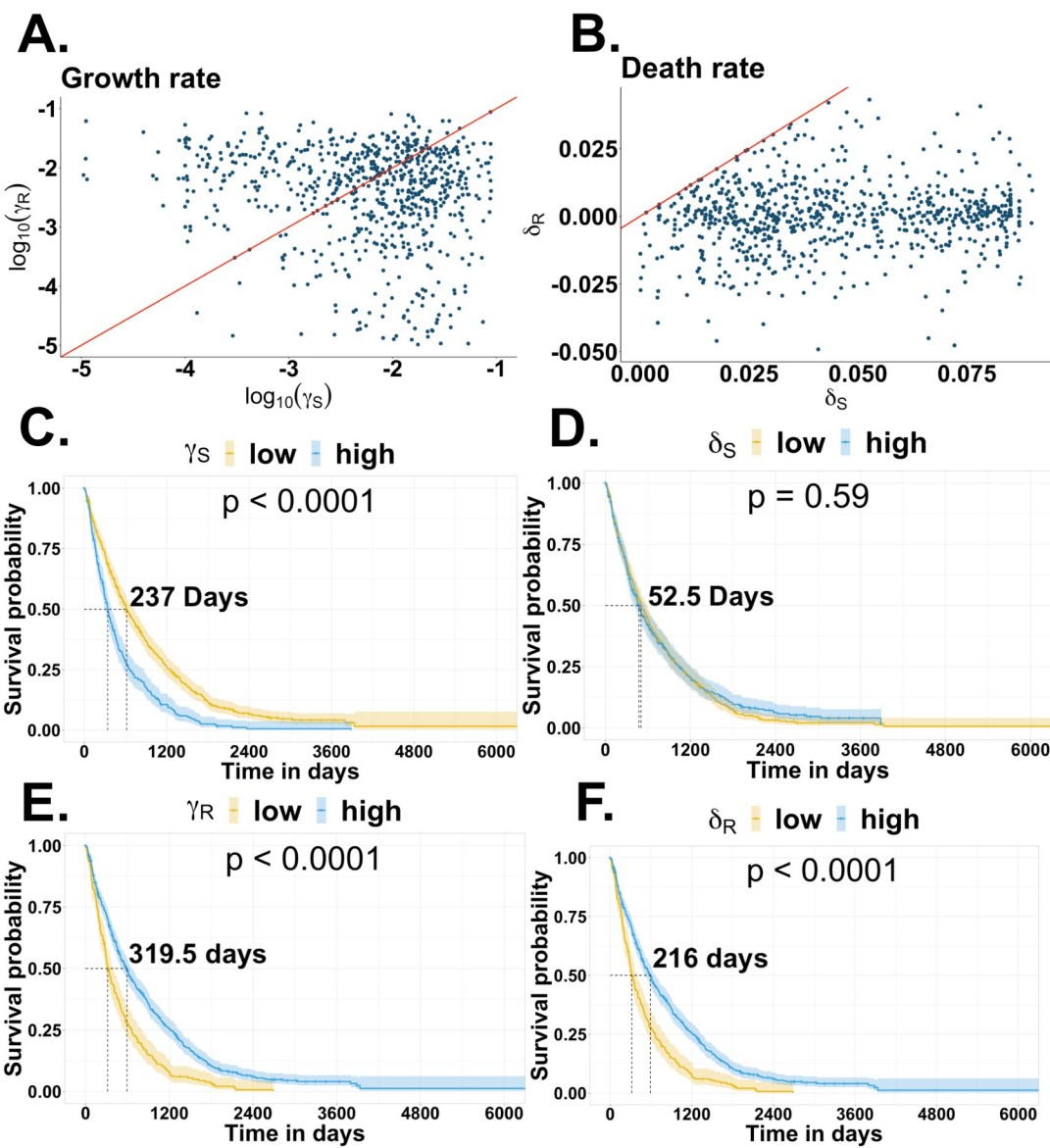

**Fig 5. Parameter fitting using *all-line fitting* from the SR and R0 model. A.** Model-estimated aggressiveness (growth rate) and **B.** Model-estimated resistance (death rate) from the SR and R0 model. The red line is the diagonal where the rates are equal. The Kaplan-Meier plot shows the survival of patients grouped by **C.** growth rate of the treatment-sensitive cell ($\gamma_S$), **D.** death rate of the treatment-sensitive cells ($\delta_S$), **E.** growth rate of the treatment-resistant cells ($\gamma_R$), and **F.** death rate of the treatment-resistant cells ($\delta_R$).

survival by 237 days), less model-estimated aggressiveness in treatment-resistant cells (small $\gamma_R$ increased median survival by 319.5 days), or lower model-estimated resistance in treatment-resistant cells (larger $\delta_R$ increased median survival by 216 days), with no effect of model-estimated resistance in treatment-sensitive cells ($\delta_S$), (Figs 5C, 5D, 5E, 5F and 6).

Next, we analyzed multivariate Cox regression by including the estimated parameters and clinical covariates. The survival outcomes after the second line of therapy were significantly influenced by multiple factors, including parameters estimated from mathematical models and clinical covariates. The resistance observed in the first-line therapy from clinical data, the post-

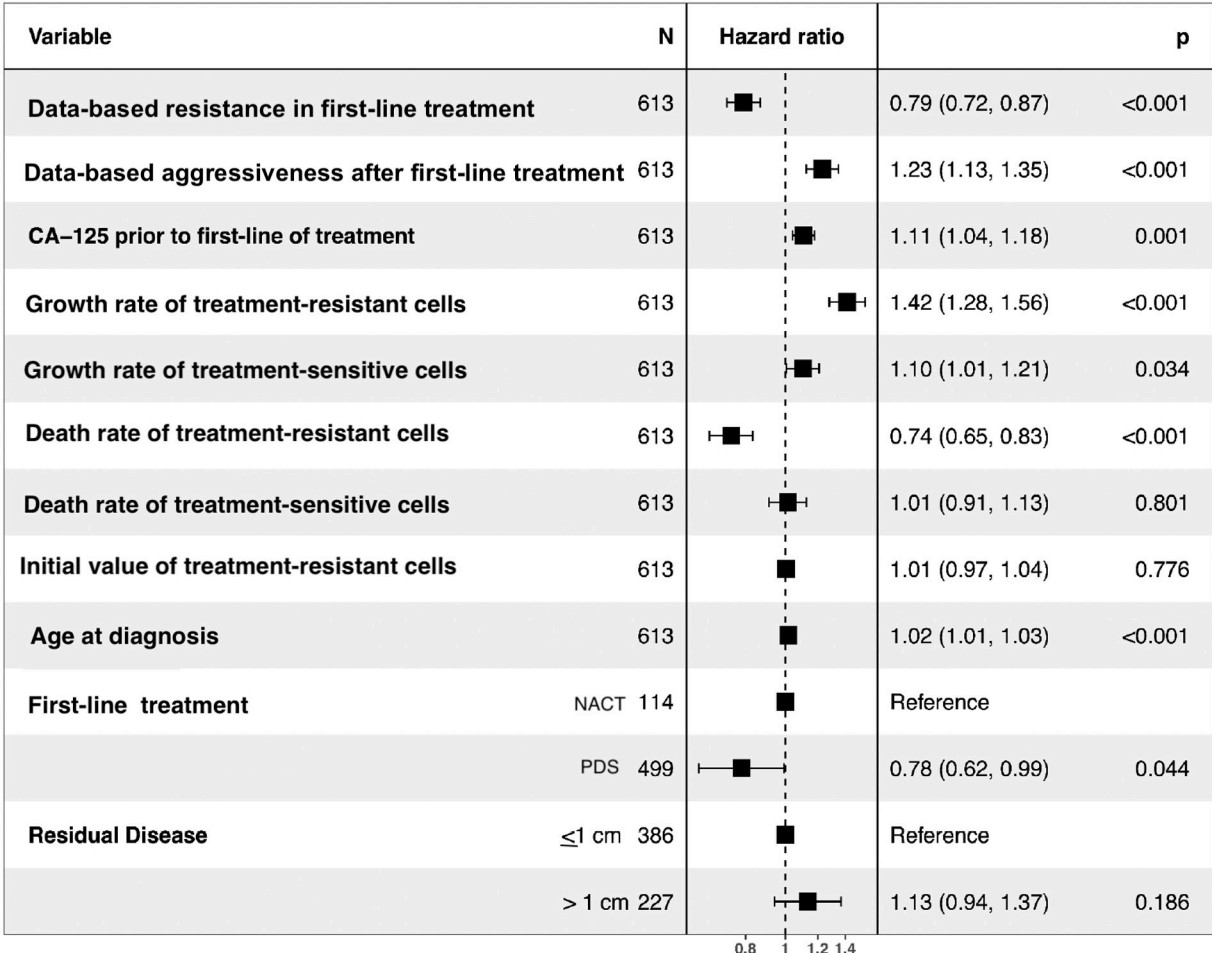

| Variable | | N | Hazard ratio | Hazard ratio (95% CI) | p |
|---|---|---|---|---|---|
| **Data-based resistance in first-line treatment** | | 613 | | 0.79 (0.72, 0.87) | <0.001 |
| **Data-based aggressiveness after first-line treatment** | | 613 | | 1.23 (1.13, 1.35) | <0.001 |
| **CA–125 prior to first-line of treatment** | | 613 | | 1.11 (1.04, 1.18) | 0.001 |
| **Growth rate of treatment-resistant cells** | | 613 | | 1.42 (1.28, 1.56) | <0.001 |
| **Growth rate of treatment-sensitive cells** | | 613 | | 1.10 (1.01, 1.21) | 0.034 |
| **Death rate of treatment-resistant cells** | | 613 | | 0.74 (0.65, 0.83) | <0.001 |
| **Death rate of treatment-sensitive cells** | | 613 | | 1.01 (0.91, 1.13) | 0.801 |
| **Initial value of treatment-resistant cells** | | 613 | | 1.01 (0.97, 1.04) | 0.776 |
| **Age at diagnosis** | | 613 | | 1.02 (1.01, 1.03) | <0.001 |
| **First-line  treatment** | NACT | 114 | | Reference | |
| | PDS | 499 | | 0.78 (0.62, 0.99) | 0.044 |
| **Residual Disease** | ≤1 cm | 386 | | Reference | |
| | > 1 cm | 227 | | 1.13 (0.94, 1.37) | 0.186 |

0.8   1   1.2 1.4

**Fig 6. The Cox proportional analysis of parameter fitting using *all-line fitting* from the SR and R0 model.** Forest plot of adjusted hazard ratios (HRs) with 95% confidence interval (CI) of hazard ratios of survival time from the end of second-line treatment to the date of last follow-up or mortality using multivariate Cox regression analysis. The death and growth rates of sensitive and treatment-resistant cells were estimated using all-line fitting with the SR and R0 model. *N* represents the number of patients used in the multivariate Cox regression. All patients with missing data for any variable were excluded from the analysis. The PDS in first-line treatment includes neoadjuvant chemotherapy (NACT) and primary debulking surgery (PDS).

first-line therapy aggressiveness based on clinical data, the pre-treatment levels of CA-125 measured from clinical data, the growth rates of treatment-resistant ($\gamma_R$) and treatment-sensitive cells ($\gamma_S$), the death rate of treatment-resistant cells ($\delta_R$), age at diagnosis, and the specific type of first-line treatment, all exhibited statistically significant impacts on patient survival after finishing the second line of therapy (Fig 6). Patients with a higher estimated death rate of treatment-resistant cells have higher survivorship (Fig 6 and Hazard ratio = 0.74). The initial population of treatment-resistant cells estimated from the mathematical model, ($R(0)$ showed no statistically significant impact on survivorship.

**Fitting of mathematical models to clinical data using *two-line fitting*.** Given that the multiple-cell type models (the SR and R0 model, the SR and $\mu$ model, and the AD model) accurately capture the entire CA-125 data set and provide parameter estimates that predict survivorship, we tested whether parameters estimated with *two-line fitting*, *three-line fitting*, and *four-line fitting* could predict the trajectory of the future dynamic of CA-125 levels in the short and long run. The model accurately fits these subsets of the data (Fig 7A). However, the model

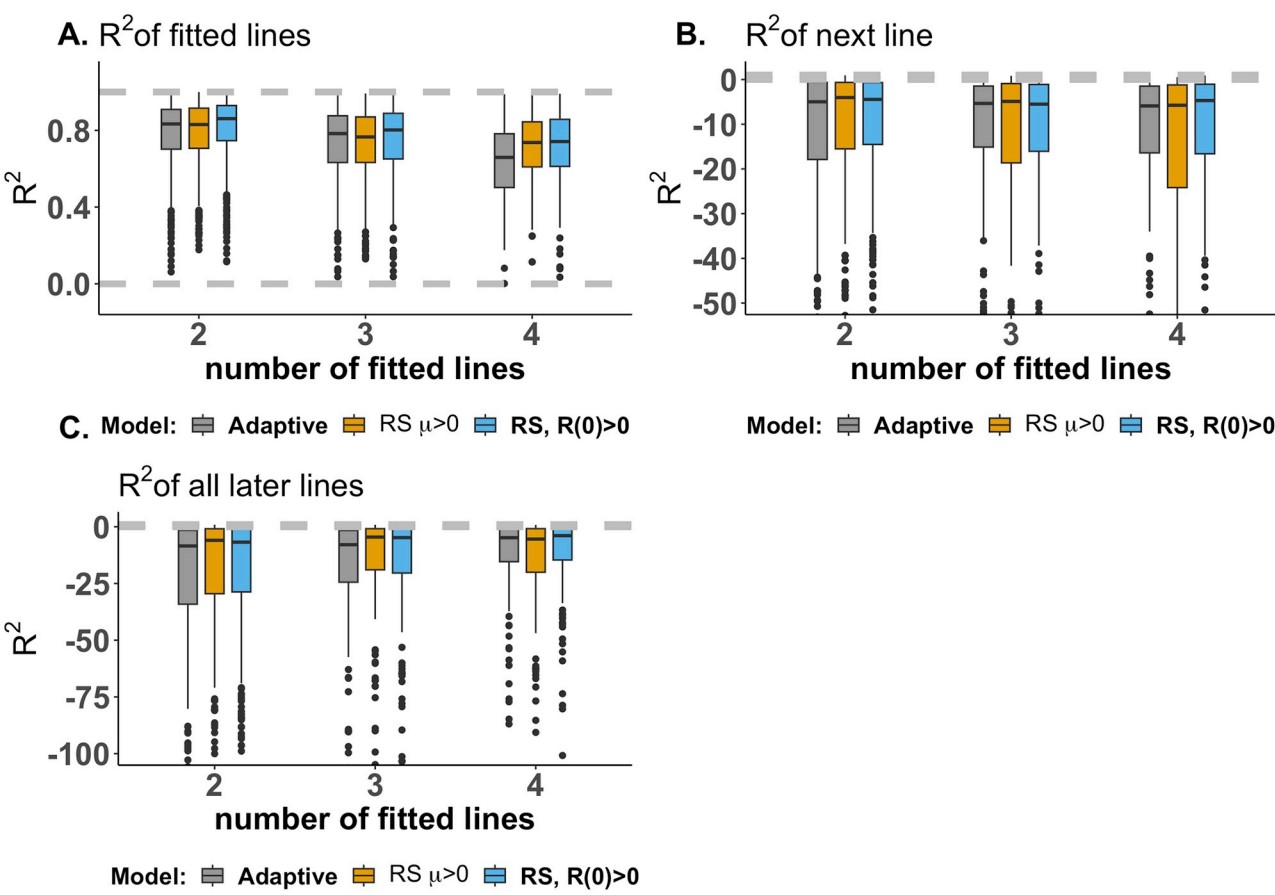

**Fig 7. Model fitting with the SR and R0 model.** Box plot showing $R^2$ of predictions of **A.** the data used for model fitting **B.** the next line after the last fitted line and **C.** all the lines after the last fitted line using *two-line fitting*, *three-line fitting*, and *four-line fitting*.

fails to accurately predict future data. Specifically, the coefficient of determination ($R^2$) for the subsequent treatment line, referred to as "R-squared next line" (Fig 7**B**), as well as for all subsequent data points that were not used for model fitting, referred to as "R-squared all later lines", were negative for all three models (Fig 7**C**), although they improve as we progressively include a greater number of treatment lines in the parameter estimation (Fig 7**B** and 7**C**).

Fitting only the first two lines predicted survivorship, with better survival for patients with low $\gamma_R$, low $\gamma_S$, and high $\delta_R$ ($p < 0.001$ and Fig 8). Reduced aggressiveness of both sensitive and resistant cells and a low level of resistance in resistant cells contributes to enhanced patient outcomes. As with *all-line fitting*, $\delta_S$ has no statistically significant effect on survival outcomes. The difference in the median survival between patients with high and low values of these parameters was similar to those found by *all-line fitting*. By considering the clinical covariates from the data with the estimated parameters from the mathematical models, we found that patients with a low growth rate of treatment-resistant cells and a higher death rate of treatment-resistant cells had a better chance of survival (Fig 9). The results from *three-line fitting* and *four-line fitting* provide similar results (S1 File).

## Discussion

This study used statistical analysis and mathematical modeling to investigate the dynamics of CA-125 levels in patients with high-grade serous ovarian cancer (HGSOC) from the Australian

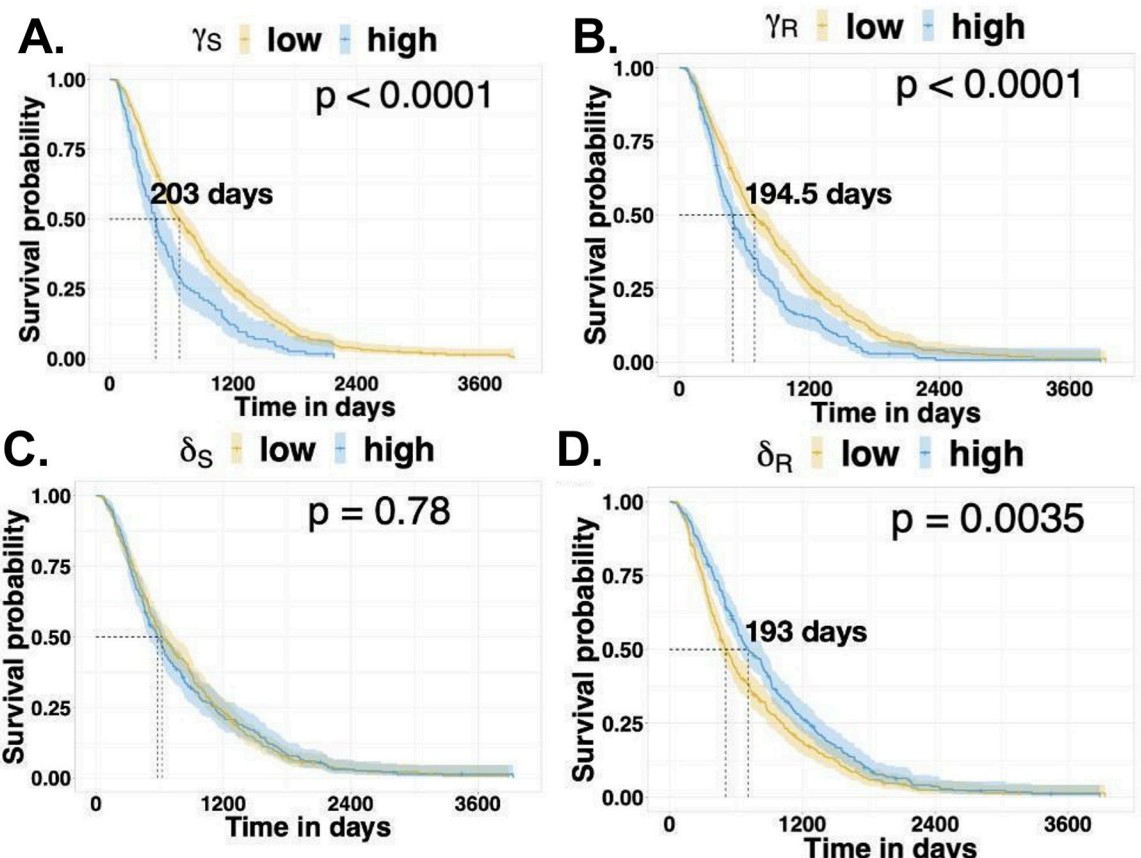

**Fig 8. Parameter fitting using *two-line fitting* with the SR and R0 model.** Kaplan-Meier plot showing the survival probabilities of HGSOC patients with low or high **A.** $\gamma_S$, **B.** $\gamma_R$, **C.** $\delta_S$, and **D.** $\delta_R$ estimated from the SR and R0 model with two-line fitting.

Ovarian Cancer Study. We found that the decline in CA-125 levels during treatment slowed with additional lines of treatment. This suggests the emergence of chemoresistance, reduced responsiveness of cancer cells to chemotherapy over time. In order to investigate the evolution of chemoresistance, we formulated and fitted mathematical models of the dynamics of CA-125 and predict patient survival.

We addressed four main issues. First, we found that simple models with two cancer cell types and two traits, *resistance* (rate of decline during therapy) and *aggressiveness* (rate of growth between lines of therapy), could successfully capture the dynamics of CA-125 levels in most HGSOC patients. By considering the interplay between resistance and aggressiveness, these models provide insight into the dynamic of cancer cells during and between lines of therapy. Models where treatment-resistant cells existed before the beginning of the treatment and those where treatment-resistant cells emerged over time fit the data equally well. Second, a model using adaptive dynamics, in which the cell state varies along a continuum, fits the data as well as ordinary differential equations with two fixed sensitive and treatment-resistant cell populations. Third, statistical analysis showed that patients with good surgical outcomes, a rapid decline in CA-125 levels during first-line therapy, and slow growth after first-line therapy had the best survival. The models predict survival and significantly improve upon survivorship models that use only empirical data. In particular, when using information from the first two lines of treatment, patients with a low estimated growth rate of treatment-resistant

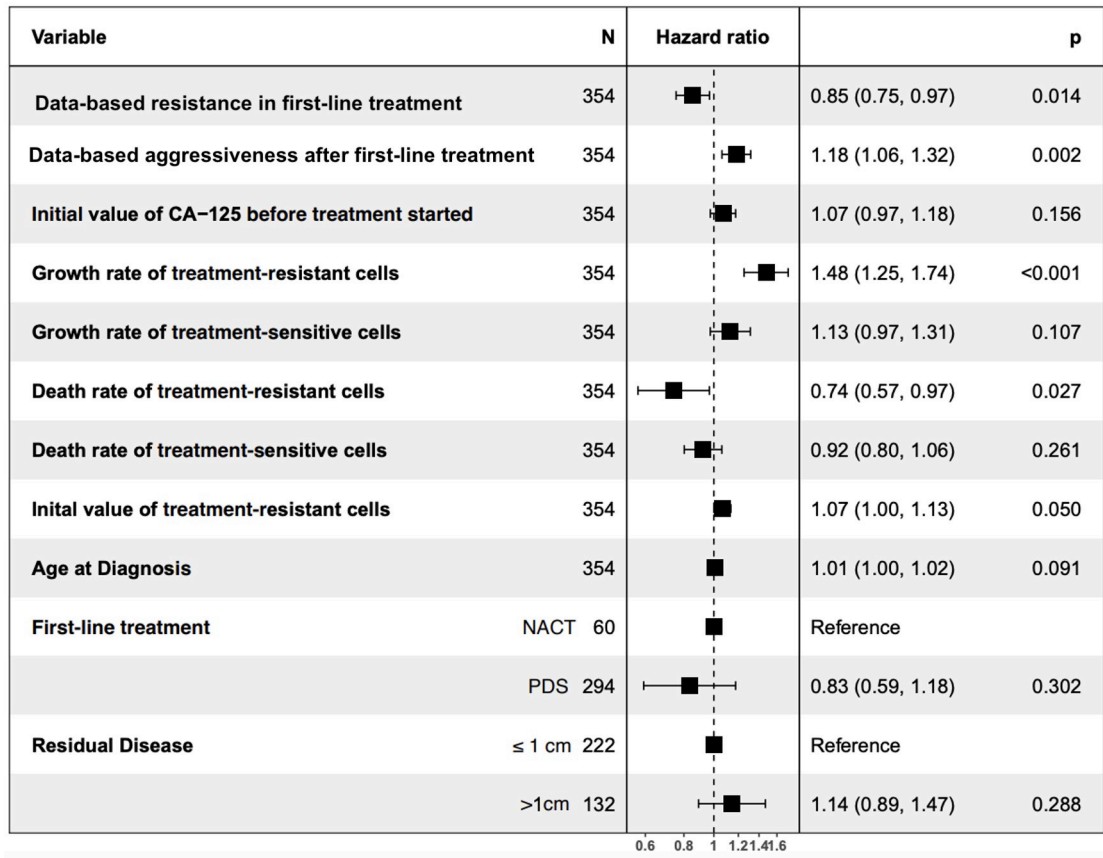

| Variable | N | Hazard ratio | | p |
|---|---|---|---|---|
| **Data-based resistance in first-line treatment** | 354 | | 0.85 (0.75, 0.97) | 0.014 |
| **Data-based aggressiveness after first-line treatment** | 354 | | 1.18 (1.06, 1.32) | 0.002 |
| **Initial value of CA−125 before treatment started** | 354 | | 1.07 (0.97, 1.18) | 0.156 |
| **Growth rate of treatment-resistant cells** | 354 | | 1.48 (1.25, 1.74) | <0.001 |
| **Growth rate of treatment-sensitive cells** | 354 | | 1.13 (0.97, 1.31) | 0.107 |
| **Death rate of treatment-resistant cells** | 354 | | 0.74 (0.57, 0.97) | 0.027 |
| **Death rate of treatment-sensitive cells** | 354 | | 0.92 (0.80, 1.06) | 0.261 |
| **Inital value of treatment-resistant cells** | 354 | | 1.07 (1.00, 1.13) | 0.050 |
| **Age at Diagnosis** | 354 | | 1.01 (1.00, 1.02) | 0.091 |
| **First-line treatment** | NACT 60 | | Reference | |
| | PDS 294 | | 0.83 (0.59, 1.18) | 0.302 |
| **Residual Disease** | ≤ 1 cm 222 | | Reference | |
| | >1cm 132 | | 1.14 (0.89, 1.47) | 0.288 |

0.6  0.8  1  1.2 1.4 1.6

**Fig 9. Cox proportional analysis of parameter fitting using *two-line fitting* with the SR and R0 model.** Forest plot of adjusted hazard ratios (HRs) with 95% confidence interval (CI) of survival time after the second-line treatment by using multivariate Cox regression analysis. The death and growth rates are estimated by *two-line fitting* from the SR model and R0 model. The data-based resistance in the first line of treatment and the data-based aggressiveness after the first line of treatment were found from the clinical data. The initial CA-125 is the CA-125 level before first-line treatment. The first-line treatment includes neoadjuvant chemotherapy (NACT) and primary debulking surgery (PDS). Age at Diagnosis, the treatment in the first line, and residual disease can be found in the data set. N represents the number of patients used in the multivariate Cox regression. Patients with any missing variables were excluded from the analysis.

cells survived on average 342 days longer than patients with a high estimated growth rate. Similarly, patients with a low estimated growth rate of treatment-sensitive cells survived on average 271 days longer than patients with a high estimated growth rate. Patients with high estimated rates of treatment-resistant cell death have a significant median survival of 262 days more than patients with low estimated death rates for these treatment-resistant cells. This finding implies a pivotal role of treatment-resistant cell characteristics for patient outcomes. The inferred death rate of the treatment-sensitive cells during treatment did not predict survival. Although, our models accurately fit the data and predict survivorship. the models fail to predict the future dynamics of CA-125 levels, both in the short term and over an extended time frame, whether based on *two-line fitting*, *three-line fitting*, or *four-line fitting*. Although the models performs better when initially fitted with more lines of treatment, the predictive power never exceeds that of a null model. Practically, this means that simple models can characterize the status of resistance in individual patients even though they cannot accurately predict the future trajectory of disease.

Our mathematical models do not include multiple factors that affect the dynamics of CA-125 and the long-term survival of HGSOC patients, such as primary surgery, residual disease after primary surgery, and the normalization of the CA-125 levels. Future studies are need to show whether additional variables can improve predictions of future CA-125 levels and the survival outcomes of HGSOC patients. According to our statistical analysis, these factors affect overall patient survival and CA-125 levels before the next line of treatment. We also did not include the effects of phenotypic plasticity [8], which could provide a much more rapid way for tumors to develop chemoresistance.

The current models do not incorporate differences among therapeutic agents used to treat HGSOC, and thus effectively assume complete cross-resistance. The most common drugs used in this study included the platinum drugs carboplatin and cisplatin, and a range of other agents, including paclitaxel, gemcitabine, liposomal doxorubicin, topotecan, bevacizumab, cyclophosphamide, and PARP inhibitors. Each type of drug has a distinct mechanism of action. Future work will include the specific drug types, the use of drug combinations, and the possibility of partial cross-resistance. This expanded scope of investigation aims to provide a more comprehensive understanding of the complexities of CA-125 level surrounding HGSOC treatment and to inform more precise and effective therapeutic approaches in the future.

Our models reveal differences in *resistance* and *aggressiveness* across patients with HGSOC. By linking these phenotypes with genetic data on individuals, these mathematical models will help to identify the mechanisms underlying these differences and provide targets for treatment. In this intricate web of genetic variations and clinical parameters, the models serve as our guiding compass, helping us navigate the treacherous terrain of HGSOC. They allow us to explore the intricate interplay between genetic mutations, signaling pathways, and the tumor microenvironment, unraveling the complex mechanisms that underlie resistance and aggressiveness. These insights become the foundation for designing targeted therapies that aim to disrupt the processes that sustain resistance or curb the aggressive growth of the tumor.

## Supporting information

**S1 Fig. Patients in the study.** The qualification of patients in the study. The diagram summarizes the inclusion and exclusion criteria in the study.
(TIF)

**S2 Fig. Parameter fitting.** Scatter plot of log-transformed CA-125 level with time (days). A graphical representation of the data used for the model fitting of **A.** *two-lines fitting* **B.** *three-lines fitting* **C.** *four-lines fitting*. Data utilized in calculating the $R^2$ value for all unfitted lines ($R^2$ all later lines) and the $R^2$ value for the subsequent line ($R^2$ the next line). Each individual vertical line denotes the date on which the patient received treatment doses, which is Cyle of treatment. The group of cycles together with gap between them is Line of treatment. The different colors of the vertical lines indicate the use of distinct drugs in each treatment line. The first-line, second-line, third-line, fourth, and fifth-line indicate the line of treatment. The indicator variable is $\alpha = 1$ during lines of treatment and $\alpha = 0$ otherwise.
(TIF)

**S3 Fig. Kaplan-Meier plots of data-based resistant and data-based aggressiveness.** Kaplan-Meier plots of survival after finishing the second line of treatment. **A.** Effect of data-based resistance during first-line treatment. **B.** Effect of data-based aggressiveness after first-line treatment.
(TIF)

**S4 Fig. Example of CA-125 level.** Examples of patients with the three patterns of CA-125 dynamics. The lines illustrate model fits using all data points for the three models: the SR and R0 model (orange), the SR and $\mu$ model (green), the single-cell model (yellow), and the adaptive dynamics model (red).
(TIFF)

**S5 Fig. Model comparison in estimation of initial condition $C(0)$. A.** Box plot of the ratio of the sum of the squared errors from the Friedman's SuperSmoother function to the sum of the squared errors from the mathematical models. **B.** Box plot of goodness of fit ($R^2$) using *all-lines fitting*. **C.** The Akaike Information Criterion (AIC) of the four models. The red dot indicates the median value of each variables. Box plot showing $R^2$ of predictions of **D.** the data used for model fitting **E.** the next line after the last fitted line and **F.** all the lines after the last fitted line using *two-line fitting*, *three-line fitting*, and *four-line fitting*.
(TIFF)

**S6 Fig. The correlation plot of estimated parameter.** Pairwise correlation comparison of estimated parameters from mathematical models using *all-line fitting* with the scatter plots, histogram representing distribution, and the correlation coefficient **A.** the SR and R0 model **B.** the SR and $\mu$ model, and **C.** the Adp model.
(TIFF)

**S1 Table. Correlation of level of data-based Resistance.** Summary and correlations of the data-based resistance estimated from the first four lines of therapy. Negative values indicate a decrease in CA-125, and positive values show an increase in CA-125. The log(CA-125) in the last column is the level of CA-125 before the patients are given therapy. *Italics* indicates significance at $p < 0.05$ and **bold** $p < 0.0001$.
(PDF)

**S2 Table. Correlation of level of data-based Aggressiveness.** Summary and correlations of the data-based aggressiveness estimated from the first four lines of therapy. *Italics* indicates significance at $p < 0.05$ and **bold** $p < 0.0001$.
(PDF)

**S1 File. Additional results.** Additional results that are not present in the main paper include the fitting from the Adaptive Dynamic model and the SR model with $\mu$.
(PDF)

## Acknowledgments

The authors would like to thank Andrea Bild, Jennifer A. Doherty, and Jon Seger for valuable conversation related to this paper.

## Author Contributions

**Conceptualization:** Kanyarat Jitmana, Jason I. Griffiths, David Bowtell, Frederick R. Adler.

**Data curation:** Sian Fereday, Anna DeFazio, David Bowtell, Frederick R. Adler.

**Formal analysis:** Kanyarat Jitmana, Frederick R. Adler.

**Funding acquisition:** David Bowtell, Frederick R. Adler.

**Investigation:** Kanyarat Jitmana, Sian Fereday, Anna DeFazio, David Bowtell, Frederick R. Adler.

**Methodology:** Kanyarat Jitmana, Frederick R. Adler.

**Project administration:** David Bowtell, Frederick R. Adler.

**Software:** Kanyarat Jitmana, Jason I. Griffiths, Frederick R. Adler.

**Supervision:** Jason I. Griffiths, Frederick R. Adler.

**Visualization:** Kanyarat Jitmana.

**Writing – original draft:** Kanyarat Jitmana.

**Writing – review & editing:** Jason I. Griffiths, Sian Fereday, Anna DeFazio, David Bowtell, Frederick R. Adler.

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
