## [Decision Letter · Decision Letter 0]

19 Oct 2023

Dear Dr. Adler,

Thank you very much for submitting your manuscript "Mathematical modeling of the evolution of resistance and aggressiveness of high-grade serous ovarian cancer from patient CA-125 time series." for consideration at PLOS Computational Biology.

As with all papers reviewed by the journal, your manuscript was reviewed by members of the editorial board and by several independent reviewers. In light of the reviews (below this email), we would like to invite the resubmission of a significantly-revised version that takes into account the reviewers' comments.

We cannot make any decision about publication until we have seen the revised manuscript and your response to the reviewers' comments. Your revised manuscript is also likely to be sent to reviewers for further evaluation.

Sincerely,

Dominik Wodarz

Academic Editor

PLOS Computational Biology

Jason Haugh

Section Editor

PLOS Computational Biology

Reviewer's Responses to Questions

**Comments to the Authors:**

Reviewer #1: Attached.

Reviewer #2: • I can’t find the units of growth/death rate – have these been reported? I would appreciate a table of parameters (units, values, brief description) for the ODE models.

• Authors appear to assume exponential growth in both ODE models (e.q's 3-5), could authors justify this choice or else check results are not affected by using another model (e.g. logistic). As a general point, it is not clear whether the authors have used existing models (i.e. from citations 32,35,36,38), whether existing models have been modified, or if the models presented here are entirely new. I would like to see more explicit comparison of similarities/differences with existing models and justification of any differences.

• Both AD and SR models assume zero tumor growth during treatment, which seems unrealistic (especially for resistant cells). Indeed in Fig. S4B it does look like tumors progress during treatment intervals. Given the large data-set, it should be possible to estimate the parameter alpha directly from the data.

• I can find no legend/description of the vertical colored bars in Figs 1, S2, & S4.

• I would like to see some discussion about what is considered to be “on treatment” in the model. As an example, in Fig. S4B there are vertical blue bars which I presume represent treatment intervals. In the model is it assumed that patients are “off treatment” during the small intervals between these blue bars, or is the patient considered to be “on treatment” throughout? What about S4B yellow bars, where the time intervals are much longer?

• I am not convinced about the robustness of the authors parameter estimation procedure. E.g. Fig. S4C single cell model fit looks very poor, it seems the data could be fit much better with a horizontal line (which the model structure does not preclude). Evidence that authors had taken steps to find global optima during parameter estimation (e.g. by performing repeated fits with different initial parameter guesses) would help assuage these doubts.

• On line 312 authors write “There was no relationship between the estimated growth or death rates of sensitive and resistant cells”. This would be an interesting finding since the presence of a fitness cost of resistance is an open question with therapeutic relevance (see e.g. Gatenby & Brown, 2018). Moreover it does look like there might be a negative correlation in Fig. 5A. Authors should state which correlation coefficient they have used – would different coefficients give different results?

• Authors model selection approach is non-standard & doesn’t take into account the number of estimated parameters. Why not use e.g. AIC/BIC? Moreover it is not clear why one particular model variant was chosen for Figs 6-8 and the others are relegated to the supplement.

• In the supplementary file a figure reference appears to be missing “(the result is presented in the paper, Fig. ??)”.

• Authors concede in the discussion that their model neglects differences between different types of therapeutic agent. It would be useful to have more information about what treatments patients received (a Sankey diagram might be a good visual representation of this). If there are some treatment combinations with enough patients, it would be interesting to see how accuracy of model predictions for future response depend on treatment combination.

Reviewer #3: Overview

In this work, the authors evaluate and investigate chemoresistance in ovarian cancer patients and predict treatment effectiveness based on individual characteristics analysed from the data (using a mathematical model of log-slope).

Cancer Antigen 125 (CA-125) levels were measured in 791 patient with high-grade serous ovarian cancer (HGSOC). They evaluated the effectiveness of different mathematical models of treatment sensitive and treatment resistant cancer cells (ODEs, AD) and fit to the HGSOC data and captured the CA-125 level. They Found that patients with log growth rates of treatment-sensitive cells and treatment-resistant cells and a high death rate of treatment resistant cells have longer survival

I have a few comments below around clarity of some explanations and suggestions for delving further into the data/model predictions. Overall, I think this is an interesting study.

Major

• The authors mention that they calculate the “data-based aggressiveness as the slope of the log-transformed CA-125 level changing when patients were not receiving therapy”

o can the authors comment on how this (and the data-based resistance measure) were confirmed to be a good measure to quantify the data?

o In between treatment lines it seems for the data shown that there is a decrease then increase, how is this captured in the single measurement? If the slope changes between the lines of treatment, how is this captured?

o I’d be curious to see what other measures might provide more or different information than what is captured in Fig 2. It seems there are very long tails for the predicted slopes, why is this?

• Can the authors comment on the reasons for delta_R not constrained to being positive? In theory it means when the drug is on, we can have growth of resistant cells be represented by this death term. I think some comment around this is necessary for the reader.

• In the data in Fig 1E, it is clear there can be a decline after treatment line ends and then eventually an increase. In the model, treatment on or off is controlled by alpha = 0 or 1. This means when the treatment line ends that alpha = 0 and we should only have growth of both population (as the delta terms should be zero) but the data doesn’t seem to depict this, can the authors explain whether this means Eq (3) and (4) don’t match data that looks like Fig 2? Or what might be causing a decline after treatment line ends that is captured in their model.

• I think the correlations across patients is very interesting and could be dug into further and moved into the main text (Table S1). e.g.

o does the minimum CA-125 between lines correlate to longer term survival?

o if you only estimate the slope post minimum between lines, does this provide a better indicator of resistance?

o Does aggressive ness and resistance at each line correlate for individual patients.

o Is there a correlation between parameter values or data values and the style of treatment (length of line? Or treatment type?)

• It would be interesting to see whether of the “three general dynamical patterns” one model fits one better, instead of just looking at all data fits per model today (Fig 4)

Minor

• I’m not sure if this is a typo but I’m unfamiliar with the name of a study being in the author list: “Australian Ovarian Cancer Study”, should that be in the author list?

• The introduction feels lacking in references to previous mathematical works modelling ovarian cancer resistance. I encourage the authors to make a small addition to the introduction mentioning previous models of Ovarian cancer treatment or of cancer resistance modelling. It allows the readers to contextualise this work and understand it’s advantages over previous models that may be available. I see they add some references, but the discussion around these and the breadth of these references feels lacking

• The figures are quite pixelated in the version I’m seeing – could this be fixed for publication

• Can more information be given on the treatments, the authors mention “first-line”, “second-line” etc but it’s not explained what this means, in particular:

o Was each first, second etc line treatment the same for each patient

o How long are these line treatments? E.g. one a day for 7 days? How heterogeneous is this across the cohort of patients?

• Can the authors explain their mixed-effects linear regression fitting process in more detail? They reference using R, can the code please be provided (even for a synthetic dataset). They explain a least-squares fitting more in detail but mention the mixed-effects model earlier.

• Typo in Fig S4 caption “/mu”

• Can the corresponding pair-parameter plots to Fig 5A and B be given to see what the other pair correlations are (if any)

• How was c1 and c2 in (1) estimated? How do they feed into the model?

• I find the reference to the three types of models a bit confusing, while there is a paragraph explaining it on the bottom of Pg 17 it would be helpful to have a diagram or shorter names. In addition the caption to Fig 4 should explain the meaning of the horizontal axis model names again for the reader.

• Sometimes the authors say “Supplementary Fig. 4” and sometimes just S4, I’d be consistent throughout

**Have the authors made all data and (if applicable) computational code underlying the findings in their manuscript fully available?**

Reviewer #1: None

Reviewer #2: **No: **Code and data is not available as supporting information or in a public repository. In particular, I believe the underlying dataset analyzed here would be of interest to the mathematical modeling community and should be made available as a supplemental file or in a public repository before publication.

Reviewer #3: **No: **Data/code is not currently available in a public repository

PLOS authors have the option to publish the peer review history of their article (what does this mean?). If published, this will include your full peer review and any attached files.

Reviewer #1: No

Reviewer #2: **Yes: **Richard J Beck

Reviewer #3: No
---

## [Decision Letter · Decision Letter 1]

1 Feb 2024

Dear Dr. Adler,

Thank you very much for submitting your manuscript "Mathematical modeling of the evolution of resistance and aggressiveness of high-grade serous ovarian cancer from patient CA-125 time series." for consideration at PLOS Computational Biology.

As with all papers reviewed by the journal, your manuscript was reviewed by members of the editorial board and by several independent reviewers. In light of the reviews (below this email), we would like to invite the resubmission of a significantly-revised version that takes into account the reviewers' comments.

We cannot make any decision about publication until we have seen the revised manuscript and your response to the reviewers' comments. Your revised manuscript is also likely to be sent to reviewers for further evaluation.

Sincerely,

Dominik Wodarz

Academic Editor

PLOS Computational Biology

Jason Haugh

Section Editor

PLOS Computational Biology

Reviewer's Responses to Questions

**Comments to the Authors:**

Reviewer #1: Thank you for your responses to my concerns. While the paper has significantly improved, there remain issues regarding the robustness of the model fitting process.

Specifically, there are conflicting points in the text regarding the model fits to the data. Lines 239-240 state that when the “ratio approaches 1, the model aligns closely with the observed data.” However, the ratios shown in Figure 4 show that they are closer to 0.5, meaning that the RSS_model is higher than the RSS_sup. In fact, lines 344-345 state that the ratios (for all but the single state model) are “all significantly less than 1,” which would imply that the models aren’t able to describe the data. The fits shown in Figure S4 also lend to this point, especially in the long term in panel B. Panel C of Figure S4 also show that the SR and R0 and adaptive dynamics models cannot describe these types of CA-125 dynamics, as their curves are more or less constant despite CA-125 changing throughout the treatment cycles.

Additionally, from the distributions of the model-estimated aggressiveness and resistance parameters shown in Figure 5, it appears that the aggressiveness and/or resistance parameters are 0 for many patients. This implies that the optimizer did not converge and instead went to the lower bound. That is likely the case for the fits shown in Figure S4C. If both aggressiveness and resistance rates are 0, then CA-125 might initially drop during the first treatment cycle, but then remain constant once treatment is turned off (which is what we’re seeing in Figure S4C). This lends to the idea that the model is not appropriate for these types of patient data. A global optimization needs to be done to ensure that the models are not collapsing at their bounds.

Finally, please include the goodness of fit values (or R^2) for each model for the fits shown in Fig S4. (And maybe even consider including different patients, as the current fits do not describe the data very well.) The R^2 values shown in Figure 7A for 4 lines of therapy are more convincing that the models can accurately describe the data. I’d be curious to see the R^2 values over all lines of the data.

Reviewer #2: Modifications in the revised manuscript have increased clarity and improved the manuscript overall. However, my concerns about the parameter estimation process and protocol for model comparison were not addressed. Since the estimated parameters values are important for the conclusions of the paper, steps should be taken to demonstrate the robustness of the parameter estimates.

“The model output with all parameters set to 0 would remain at the initial condition and provide a poor fit.”

This response highlights a flaw in the fitting procedure, namely that the initial level of CA-125 was apparently not included as a parameter to be estimated. Instead all fits seem to be constrained to pass through the first data point, which has the undesirable effect of overstating the importance of the first data point. The impact of this constraint on the results can most clearly be seen in figure S4B, where the gradient of the line is much steeper than it would be had the fit been unconstrained at the first timepoint.

I struggle to believe the claim (regarding the yellow line in figure S4B) that “the horizontal line does not fit better in the current model”, because the yellow line is simply so far from the data points. It could be the case that the fit appears worse than it actually is, because the fitting was performed on non log-transformed data whilst the plot displays log-transformed data (which if true, should be indicated on the plot y axis or in the legend). However even then I have difficulty imagining that a horizontal line would provide a worse fit, particularly if the initial condition is fitted as a parameter as it ought to be.

“In the present study, our primary objective is to assess and compare the predictive performance of each mathematical model rather than focusing on selecting the most appropriate model.”

Even if the goal is to compare but not select a model, common model selection criteria (AIC, cross validation) are a widely used and accepted basis for comparing models. Two criticisms of the current method of comparison are 1) does not take into account model complexity, and 2) uses the super smoother as a benchmark for model performance. The super smoother is itself a nonparametric model of the data, so it seems very strange to benchmark the parametric models against this other nonparametric model rather than simply against the data itself. If there was some compelling reason to use this approach rather than standard criteria, I cannot find it in the text. It would be helpful if the authors could point to instances in the literature where this approach has been used for model comparison before. This approach should be much better motivated or else replaced with a more suitable alternative.

Reviewer #3: Thank you for your revision of your manuscript. I have four remaining comments.

1. In my previous review (Major 1.2 and 1.3), I commented on the miss-match between the data and the model after the first and second treatment line. The authors said they attempted to fit the changing slope using a fixed delay and it made no difference to their ability to fit the model, however, this still concerns me as there is a clear decrease post first and second line treatment in Fig S2 and Fig 1A that contrasts with what is modelled. Potentially, this isn't common across the patients, and maybe that should be included in the text. Either way, I think a more detailed explanation in the text is needed about why this isn't important and the results of attempting to change the model to match it for clarity for the audience. If the details have already been added, can the line numbers be pointed out.

Have the authors examined fixing the delay to be the turning point in the patient data? I'd be interested into why this wouldn't give a better fit to the data.

2. Also, can the authors please add the details in their responses to my questions in the round 1 revision to the text. In particular,

• Comment Major 4 – can the authors add to the text their analysis of the correlations between variables and significance/non-significance of these relationships

• Comment Minor 4 – give more details of the range seen for timing and schedule of dose lines across patients

• Comment Minor 5 – add “R with lm and lme4” for transparency and reproducibility.

3. I do not understand the authors answer to Minor 7, can they please provide either further explanation of their response or re-examine my question. Pointing out line numbers would also help.

4. Having a Fig S2 and a Fig S2 in the SI is confusing. Please changing the names of the figures in the SI to something else e.g. Fig SI2.

**Have the authors made all data and (if applicable) computational code underlying the findings in their manuscript fully available?**

Reviewer #1: Yes

Reviewer #2: **No: **No software or data appears to be deposited in a repository

Reviewer #3: **No: **I asked for code to be provided in my original review. Will the code be provided?

PLOS authors have the option to publish the peer review history of their article (what does this mean?). If published, this will include your full peer review and any attached files.

Reviewer #1: No

Reviewer #2: **Yes: **Richard J Beck

Reviewer #3: No
---

## [Editor Report · Decision Letter 2]

12 Apr 2024

Dear Dr. Adler,

We are pleased to inform you that your manuscript 'Mathematical modeling of the evolution of resistance and aggressiveness of high-grade serous ovarian cancer from patient CA-125 time series.' has been provisionally accepted for publication in PLOS Computational Biology.

Best regards,

Dominik Wodarz

Academic Editor

PLOS Computational Biology

Jason Haugh

Section Editor

PLOS Computational Biology

---

## [Editor Report · Acceptance letter]

23 May 2024

PCOMPBIOL-D-23-01237R2 

Mathematical modeling of the evolution of resistance and aggressiveness of high-grade serous ovarian cancer from patient CA-125 time series.

Dear Dr Adler,

I am pleased to inform you that your manuscript has been formally accepted for publication in PLOS Computational Biology. Your manuscript is now with our production department and you will be notified of the publication date in due course.

With kind regards,

Anita Estes
